# Combination of Two Long-Acting Antipsychotics in Schizophrenia Spectrum Disorders: A Systematic Review

**DOI:** 10.3390/brainsci14050433

**Published:** 2024-04-26

**Authors:** Salvatore Cipolla, Pierluigi Catapano, Daniela D’Amico, Rocchina Monda, Nunzia Paola Sallusto, Francesco Perris, Valeria De Santis, Francesco Catapano, Mario Luciano, Andrea Fiorillo

**Affiliations:** Department of Psychiatry, University of Campania “Luigi Vanvitelli”, Largo Madonna delle Grazie 1, 80138 Naples, Italy; salvatore.cipolla@unicampania.it (S.C.); pierluigi.catapano@unicampania.it (P.C.); daniela.damico1@studenti.unicampania.it (D.D.); rocchina.monda@studenti.unicampania.it (R.M.); nunziapaola.sallusto@studenti.unicampania.it (N.P.S.); francesco.perris@unicampania.it (F.P.); valeria.desantis@unicampania.it (V.D.S.); francesco.catapano@unicampania.it (F.C.); andrea.fiorillo@unicampania.it (A.F.)

**Keywords:** schizophrenia, antipsychotics, long-acting, LAI, clozapine, treatment-resistant

## Abstract

Background: Up to 34% of patients with schizophrenia are resistant to several treatment trials. Lack of continuous and adequate treatment is associated with relapse, rehospitalization, a lower effect of antipsychotic therapy, and higher risk of side effects. Long-acting injectables antipsychotics (LAI APs) enhance compliance and improve clinical outcomes and quality of life in patients with schizophrenia, and thus it may be advisable to administer two LAI APs at the same time in cases of treatment-resistant schizophrenia. The purpose of this review is to summarize the available literature regarding the combined use of two LAI APs in patients with schizophrenia or other psychotic spectrum disorders. Methods: An extensive literature search for relevant articles regarding any combination of two long-acting injectable antipsychotics has been performed from inception up to 9 February 2024, on PubMed, Scopus and APA PsycInfo, according to the PRISMA statement. Only studies reporting combination of two LAI APs and its clinical outcome in patients with schizophrenia and related disorders were selected. Results: After the selection process, nine case reports, four case series and two observational retrospective studies were included in the final analysis. All patients treated with dual LAI APs reported a good response, and no new or unexpected adverse effects due to the combination of two LAIs were reported. Different drug combinations were used, and the most frequent association resulted in aripiprazole monohydrate + paliperidone palmitate once monthly (32 times). Conclusions: Our review highlights that the treatment regimen with two concurrent LAI APs is already widely used in clinical practice and is recognized as providing a promising, effective, and relatively safe therapeutic strategy for treating the schizophrenia spectrum disorders.

## 1. Introduction

Schizophrenia is a severe mental disorder with a prevalence of 0.36% in the general population [1], affecting approximately 1% of adults worldwide [2]. A recent study estimated that, globally, 21 million people are living with schizophrenia and this is expected to increase in the years to come, due to population ageing [3] and the younger onset age [4,5]. Schizophrenia has traditionally been associated with concepts of progression, relapse, and chronicity. Considerable longitudinal research has changed this outlook to one that is less negative, since the majority of individuals with schizophrenia have the potential to obtain a stable symptoms remission, and adequate levels of functioning [6]. However, full recovery is not always achieved, and many patients’ needs are far from being met [7,8]. One of the most accepted theoretical frameworks of personal recovery is constituted by five processes, called Connectedness, Hope and optimism about the future, Identity, Meaning in life and Empowerment (CHIME) [9]. In our opinion, without an adequate and integrated treatment for schizophrenia, recovery cannot be achieved, and after-effects will continue to fall on patients, their families, and the health system, with the costs of frequent hospitalizations, repeated therapeutic trials and unsuccessful psychosocial interventions rising higher.

Schizophrenia spectrum disorders are frequently associated with significant distress [10,11,12] and impairment of personal, family, social, educational and occupational functioning, as well as in other important areas of life [13,14,15], being responsible for 7.4% of disability-adjusted life years (DALYs) and for 7.1% of years of life lost to premature mortality (YLLs) [16,17,18]. Patients with schizophrenia have an increased prevalence of cardiovascular and metabolic diseases, such as obesity, dyslipidaemia, diabetes mellitus, and metabolic syndrome, which has resulted in a significantly lower life expectancy [19,20,21]. The adoption of unhealthy lifestyle behaviors (such as poor eating and physical activity habits, heavy smoking, alcohol or drug abuse, and reduced access to screening programs and check-up visits for physical disorders) is primarily to blame for the high rates of comorbidity between mental and physical illnesses [22,23]. This mortality gap in patients with schizophrenia and other mental disorders is further widened by inadequate treatment of chronic physical disease [18]. On the other hand, new psychosocial intervention improving physical activities and dietary habits could reduce the gap, promoting more active and healthy living [24,25].

Since their introduction in the 1950s, antipsychotics have become the first line treatment for people with schizophrenia. They can reduce the symptoms of acute psychosis and risk of relapse. Despite this, it has been reported that up to a third of patients do not respond, and recent concerns have been raised about their efficacy in long-term use [26,27]. Moreover, in light of their serious and unpleasant adverse effects, some research suggests that some people may have better overall outcomes in the long-term if they attempt to discontinue the medication and avoid or minimize long-term use [28]. It has to be said that a range of effective care options for people with schizophrenia exist, and the need for integrated therapy is widely recognized [29,30,31,32,33,34]. For patients with an acute exacerbation or recurrence of psychosis or schizophrenia, the National Institute for Health and Care Excellence (NICE) guidelines suggest the administration of one oral antipsychotic in conjunction with psychological interventions, such as family intervention and individual Cognitive Behavioural Therapy (CBT) performed by an expert therapist [35]. Psychosocial interventions, which include supported employment and/or psychoeducation, should be offered to all individuals with schizophrenia and their caregivers. Patients may also prefer to try these psychosocial interventions with no/minimal antipsychotic drugs, although with less evidence of efficacy [36,37,38]. Other, less used but still valid, non-pharmacological treatments are available for patients with refractory psychosis, such as Electroconvulsive Therapy (ECT) [39].

Nevertheless, despite the availability of effective psychological treatments and rehabilitation programs, antipsychotics still represent the main treatment strategy to provide patients with an effective symptom control, clinical stability [34], relapse control [40] and better quality of life [41], although with a marked variability between individuals [42].

Unfortunately, up to 34% of patients with schizophrenia are resistant to different treatment trials, even at a first episode of psychosis [43,44,45]. A variety of different definitions of treatment-resistant schizophrenia (TRS) have been formulated [46]. Recently, the Treatment Response and Resistance in Psychosis (TRRIP) working group was formed to establish consensus criteria to standardize the definition of TRS, which can therefore be summarized as the persistence of symptoms despite ≥2 trials of antipsychotic medications of adequate dose and duration with documented adherence [47]. The Food and Drug Administration (FDA) indicates clozapine as atypical antipsychotic drug for TRS [48]; however, up to 60% of patients do not respond to adequate treatment with clozapine [49,50], and its range of adverse effects and the need for frequent blood monitoring make compliance an issue for many patients [51,52]. However, clozapine may be useful to treat TRS but does not appear to be the most adequate drug to treat patients with pseudo resistance (i.e., those who have been misdiagnosed or patients who have compliance issues or those who have received inadequate dose, duration, or attain sub-therapeutic plasma levels of antipsychotic medication, and cases with medical or psychiatric comorbidities or side effects of medications overshadowing the response to antipsychotic medications). Poor adherence to treatment is one of the leading causes of relapse and rehospitalization for patients with schizophrenia [53]. In the Schizophrenia Outpatient Health Outcomes (SOHO) study, which involved more than 7000 patients, treatment discontinuation rates resulted between 34% and 66% over 36 months of follow-up [54]. Lack of continuous and adequate treatment is associated with increased relapses and rehospitalizations [55], a significantly lower effect of antipsychotic treatment once reintroduced [56] and higher risk of side effects.

Long acting injectable (LAI) antipsychotics (APs) are concentrated formulations which, following intramuscular injection, release the antipsychotic drug slowly over time. This allows fewer administrations (injections at intervals that range from 2 weeks up to several months), while ensuring sustained medication coverage [57]. First generation LAI APs are typically esters of the original molecule, which allows the drug to be delivered in an oily solvent. Second generation drugs use different mechanisms to allow its gradual release, such as the microsphere system or suspension of nanoparticles in aqueous solution. Several LAI APs have been approved for use in patients with schizophrenia in the U.S. and Europe [58], although the availability of these drugs is not homogeneous between different countries. Unfortunately, in many low- and middle-income countries (LMICs) antipsychotics of any class are not always available [59], and actions to prioritize access to LAIs in LMICs should be carried out [60]. This could not only improve adherence to treatments, but also overcome difficulties in offering and maintaining regular clinical follow-ups [61]. Among the first generation LAI APs, only fluphenazine decanoate and haloperidol decanoate are approved in both the U.S. and the EU, while the remaining bromperidol decanoate, flupentixol decanoate, perphenazine decanoate, pipotiazine palmitate, and zuclopenthixol decanoate are licensed across few European countries but not the U.S. With respect to second generation antipsychotics, several molecules are available as LAI, including aripiprazole, risperidone, paliperidone palmitate and olanzapine. All of them are approved by both European and American regulatory agencies.

Compared to oral medications, LAI APs have a similar pharmacodynamic profile but can be useful to increase adherence to antipsychotic treatments [62], as many studies demonstrate that long-acting injectable antipsychotics enhance compliance and improve clinical outcomes and quality of life in patients with schizophrenia [63,64,65,66]. Furthermore, LAI APs may have clinical and neurobiological benefits early in the illness, arguably due to the assured continuity of treatment [64,67].

The most ambitious schizophrenia treatment goal would be the availability of new drugs with antipsychotic proprieties and a better tolerability profile, proven efficacy on patients’ cognitive and social functioning, more stable blood concentration, and improving the quality of life [41]. Unfortunately, such an ideal drug is not yet available [68,69]. Consequently, there is a need for new strategies for the treatment of schizophrenia and related disorders, and this urgency is especially true for patients suffering from treatment-resistant schizophrenia and for those who are poor/not compliant with common oral medications. The challenges in treating schizophrenia prompted several authors to look for new treatment strategies [70,71,72,73]. A recent umbrella review analyzed 63 randomized control trials and 29 meta-analyses, focusing on the use of adjunctive agents in antipsychotics used to treat schizophrenia, such as amino acids, hormones and anti-inflammatory drugs [74].

In TRS, the administration of more than one antipsychotic is needed [75] when it is not possible to obtain adequate control of symptoms with a single drug [76], or when a lower daily dose of a drug is needed to limit its adverse effects. More rarely, the addition of another antipsychotic may be necessary to treat symptoms other than positive or negative ones, including anxiety, insomnia and aggressive/impulsive behaviors [77]. To date, the supporting evidence on the efficacy and safety of oral polypharmacy is inconsistent [76,78,79,80,81]. On the contrary, the prescription of more than one oral antipsychotic medication can greatly reduce patient adherence to treatment [80]. This leads to the conclusion that it may be desirable for some patients to administer two LAI APs at the same time, taking advantage of their synergistic effect and safer modes of administration. However, data about the efficacy of the administration of more than one LAI APs in clinical practice are scarce.

The purpose of this systematic review is to summarize the literature regarding the combined use of two long-acting antipsychotics in patients with schizophrenia or other psychotic spectrum disorders, evaluating its efficacy and safety profile in order to provide practical indications for the clinician who interfaces with treatment-resistant patients who cannot benefit from the use of clozapine.

## 2. Methods

### 2.1. Search Strategy

An extensive literature search for relevant articles has been performed from inception up to 9 February 2024 on PubMed, Scopus and APA PsycInfo. The three different search keys adopted for each online database are shown in Appendix A.

The search method has been carried out according to the Preferred Reporting Items for Systematic Review and Meta-Analysis (PRISMA) statement, as applicable [82]. The PRISMA checklist is reported in Appendix A. A protocol was not registered.

Population-Intervention-Comparison-Outcome (PICO) criteria have been established to determine eligibility for inclusion, as follows: (a) population: patients diagnosed with schizophrenia spectrum disorder and other psychotic disorders, according to DSM-5 [83], with no restrictions on the patient’s gender, ethnicity and age; (b) intervention: any combination of two long-acting injectable antipsychotics, even if dosages and frequency of administration were not specified; (c) comparison: the standard of care for schizophrenia, consisting of mono- or poly-therapy with oral antipsychotics or the combination of oral antipsychotics and one single long-acting antipsychotic; (d) at least one among the following outcomes: changes in clinical measures, the remission rate, and tolerability and rehospitalization rate. The following conditions were taken as exclusion criteria: (a) pre-clinical studies; (b) systematic reviews and meta-analysis; (c) studies in which the antipsychotic formulation is not explicitly specified; (d) articles written in a language other than English, whose data could not be obtained from other records. Finally, the reference lists of included articles were screened to identify additional relevant studies.

### 2.2. Selection Process

A total of 620 papers were identified from our search. After controlling for duplicates, 177 papers were removed. Of the remaining articles, 20 articles were excluded since they were reviews (12), comments (5) and corrigenda (3). After a full text reading, a total of 414 articles were removed, as they did not meet the inclusion criteria. Moreover, six papers were added by snowballing and citation checking [84,85,86,87,88,89]. Notably, one article was included even though it is a review [85], as it also includes relevant case reports; furthermore, one article written in German resulting from PubMed database [90] and one written in Spanish [87] were also included in the final analysis, as all the relevant data were available in English in another record [85]. Finally, N = 15 papers were included in the present review analysis. The selection process is shown in Figure 1.

Relevant data were extracted by DD, RM and NPS. SC and PC triple-checked the extracted data for accuracy. Two authors resolved disagreements through discussion or by involving a third and fourth author (ML and FP).

Inter-rater reliability, referring to the degree of agreement between researchers, has been calculated, with a Cohen’s kappa score of 0.9.

### 2.3. Risk of Bias Assessment

Two authors (SC and PC) independently evaluated each selected study for the risk of bias. The ROBINS-I (Risk Of Bias In Non-randomized Studies—of Interventions) tool was used for non-randomized studies [91], and the overall risk of bias was rated as high in the two non-randomized studies included in the review; Table 1 shows the considered domains and sub-domains. The methodological quality of the case reports and case series was independently assessed by two authors (SC and PC) using the Joanna Briggs Institute (JBI) critical appraisal checklist for case reports and case series [92,93].

## 3. Results

A total of 15 studies was included in this review. Of these, nine were case reports [84,86,87,88,89,90,96,97,98], four case series [85,99,100,101], and two were observational retrospective studies [94,95]. No randomized clinical trial has been found that met the inclusion criteria. A summary of selected clinical studies and case reports/case series is provided in Table 2 and Table 3, respectively. The majority of studies were carried out in Europe [84,87,88,90,94,96,98], three in the USA [86,89,97], and two in Australia [100,101].

Articles included a total sample of 123 patients. Of these, 29 (23.57%) were male and 11 (8.94%) female; no data about gender were available for the remaining 94 subjects (63.48%), who were all included in one study [95]. The sample’s age ranged from 16- [101] to 67- [85] years-of-age. All patients received a diagnosis of schizophrenia (N = 68) [84,85,86,87,89,94,95,96,97,98,99,100,101] or schizoaffective disorder (N = 43) [85,88,90,94,95], with the exception of 12 patients, whose main diagnosis was represented by bipolar disorder (N = 6), organic diseases (N = 2), and intellectual disability (N = 4) [95]. The duration of illness extended from two [85] to thirty-two [88] years; in two cases, patients treated with dual LAI APs were at their first episode of psychosis [101]; however, this information is not consistently reported across studies [84,89,94,95,97]. The most frequently reported comorbidities were substance use disorder (SUD) and personality disorders (PD) [85,101]. Other physical comorbidities included hyperlipidemia [86,98], obesity [86,98], hypertension [85,86], acne inversa [98], and Wolf–Parkinson–White syndrome [85].

In some articles, symptoms that present prior to the treatment with dual LAI APs have been described [84,86,88,90,96,97,98,99,101]. Symptoms reported by patients before the prescription of dual LAI APs, grouped by psychopathological dimensions, are reported in Table 4, despite this data not being reported in the majority of the sample (90%) [85,87,94,95,99,100].

In all cases, the administration of dual LAI APs was not the first pharmacological choice, since all patients underwent at least one previous trial with oral and/or long-acting Aps; olanzapine, at different formulation and dosage, was the most frequently administrated antipsychotic [84,85,87,96,97,98,99,101].

A previous trial with oral clozapine was reported in 17 patients [84,85,87,88,90,94,96,97,100]. Reasons for discontinuation were: refusal of oral medication and/or non-adherence to oral treatment in six patients [85,94,96,97,100], occurrence of adverse events such as leukocytopenia [84,87,90], sedation [84,85,94], orthostatic hypotension [84], weight gain [94], and unspecified adverse events [100]; in one case clozapine discontinuation was due to inefficacy [100] and in one report the reason was not specified [88]. In six cases clozapine was not used due to prior refusal and/or non-adherence to oral medication [85,98,101]. In 13 patients oral clozapine was not provided [85,94]. In four studies no information regarding prior prescription of clozapine was provided [86,89,95,99]. Notably, one patient described by Wartelsteiner and Hofer tried oral clozapine three times before receiving two concomitant LAI APs [84].

In Table 5 all the different combination of LAI APs used in the selected studies are listed. The concomitant usage of FGA and SGA LAI resulted in 12 different combinations; five combinations of two different SGA LAIs are described, with aripiprazole monohydrate and paliperidone palmitate being the most widely used drugs; four combinations between two FGA LAIs are described, with haloperidol decanoate, zuclopenthixol decanoate, and flupentixol decanoate being the most frequently used, each reported in five cases. The most frequently adopted combination was aripiprazole monohydrate + paliperidone palmitate once monthly (32 times; 26% of cases) [85,94,95], followed by flupentixol decanoate + paliperidone palmitate once monthly (22 times; 17.88% of cases) [85,95,99], zuclopenthixol decanoate + paliperidone palmitate once monthly (20 times; 16.26% of cases) [94,95,101], zuclopenthixol decanoate + aripiprazole monohydrate (six times; 4.87% of cases) [85,94,95], haloperidol decanoate + paliperidone palmitate once monthly (6 times; 4.87% of cases) [86,95], haloperidol decanoate + zuclopenthixol decanoate (five times; 4.06% of cases) [94,95], haloperidol decanoate + flupentixol decanoate (5 times; 4.06% of cases) [95]; other combinations were used less frequently.

Finally, the administration schedule consisted of the injection of the two drugs alternately in nine cases [88,98,99,100], and the simultaneous injection of the two LAI APs in seven cases [85,90], while the scheme was not specified for the remaining 107 patients [84,85,86,87,89,94,95,96,97,101]. Not all drug combinations were administered exclusively to patients with schizophrenia or other related disorders; for example, in the Kenar study (2023), 12 subjects were suffering from other psychiatric disorders or organic illnesses. Therefore, it was not possible to establish which treatments were intended for patients affected by schizophrenia.

Efficacy outcome was assessed mostly in terms of symptom reduction following a clinician-led evaluation, being reported in seven papers (46.66%) [84,86,87,89,90,97,99]. In other cases, the efficacy of dual LAI APs administration was assessed in terms of reduction of total number of hospitalizations and length of stay [85,94,95,100], reduction of number of referrals to the emergency department [85], and reduction of validated psychometric test scores, such as the Clinician-Rated Dimensions of Psychosis Symptom Severity (CRDPSS) [88], the Positive and Negative Symptoms Scale (PANSS) [96,98], the Brief Psychiatric Rating Scale (BPRS) [96,101], and the Clinical Global Impression (CGI) [96].

In all selected case reports, a symptom improvement or a clinical stabilization with functional recovery was described. The persistence of slight residual symptoms was highlighted in three patients [86,90,97], while in three cases it was specified that the introduction of the regimen with two LAI APs allowed the patient to be discharged from the hospital [86,88,89]. In the case reported by Scangos et al. (2016), the patient reported an improved compliance with other oral medications [97].

Legrand et al. (2014) reported an improvement of all CRDPSS scores [88]. Lenardon et al. (2017) found a 43.9% reduction at PANSS and BPRS scores, and in both CGI Illness Severity and Global Improvement scores [96]. Similar results have been reported by McInnis and Kasinathan (2019) [101], and by Jarosz and Badura-Brzoza [98].

Mathew et al. (2018) and Youykheang et al. (2023) [85,100] found reductions in the number of hospitalizations following the introduction of dual LAI APs. It should be noted that the follow-up period is extremely variable in the eight cases of Youykheang et al. (2023), extending from 12 to 45 months [85]. Accordingly, two retrospective observational studies recorded a significant decrease in the number of hospitalizations after the introduction of dual LAI APs [94,95].

The occurrence of adverse events after dual LAI APs administration was found in four patients (3.25%), including unspecified extrapyramidal symptoms [85], mild bradykinesia [101], and polydipsia and polyuria [101]. One observational retrospective study measured safety by monitoring biological variables, such as BMI, agranulocytosis, lipid profile, and sugar levels, and no significant changes were observed pre- and post-treatment [94].

## 4. Discussion

Our review highlights that, among the identified therapeutic strategies, the use of two LAI APs is rapidly growing in clinical practice, despite current guidelines not providing recommendations for this option. This happens regardless of age and illness duration, being reported both in cases of stable schizophrenia and after the first psychotic episode [101]; the presence of psychiatric and/or physical comorbidities does not influence the choice to use this regimen. Notably, obesity was present in two reports [86,98], as well as arterial hypertension [85,86] and hyperlipidemia [86,98].

Several studies proved that LAI APs have comparable or higher efficacy than the same oral molecules [63,66]. In our review all patients treated with dual LAI APs reported a good response, measured as the number and extent of symptoms. In the selected articles, an improvement in the scores of all used standardized psychopathological scales was also observed. In addition, the number and length of hospitalizations of patients taking dual LAI APs also appears to be reduced. The number of psychotic episodes and the length of previous hospitalizations correlates with the risk of further hospitalizations [102], so it is possible to expect that the use of two concurrent LAI APs may improve long-term outcomes.

Since the prior treatment regimen has not always been reported, this review does not allow us to establish with certainty whether the reduction in the hospitalization rate is due to switching from an oral drug to a long-acting drug or the concomitant use of two LAI APs. The efficacy of dual LAI APs regimen is also confirmed for patients who have made unsuccessful attempts with oral drugs, including clozapine. The combination of oral clozapine and LAI APs is also possible in selected cases of TRS [103].

In addition, with clozapine the problem of non-adherence and the need for periodic monitoring of plasma levels and the appearance of granulocytopenia remains. LAI APs appear to be effective in reducing hospital readmissions in patients with psychosis with a history of poor treatment adherence [104], and increased adherence reduces the relapse risk [105]. In one case, dual LAI APs therapy was shown to be effective where non-pharmacological treatments—such as electroconvulsive therapy (ECT)—had also failed [100].

According to our data, the two intramuscular injections can be carried out simultaneously or alternately on a weekly/bi-weekly basis, and this choice does not seem to have consequences for the effectiveness of the treatment. In addition, simultaneous or alternating administration of LAI APs does not appear to modify the occurrence of adverse events. Clinicians administering LAI APs should consider the pharmacokinetic properties of these molecules to maximize their clinical impact [106], avoiding concentration peak in the hours following administration, and thus limiting excessive receptor occupancy and the consequent onset of adverse effects, such as extra-pyramidal symptoms, sedation, and hyperprolactinemia. In addition, since the IM puncture can cause pain or injury at the injection site, it would be desirable to alternate the muscle with each drug administration.

We were not able to identify a specific prescription pattern when combining two different LAI APs. In the majority of cases, the second LAI AP to be administered was chosen based on the partial response received from the administration of the analogous oral molecule. However, the reason for the specific combination of two LAI APs chosen is often random or unspecified. Some authors suggest the combination of first- + second-generation antipsychotic drugs to exploit the synergistic effect of the joint administration of the two long-acting antipsychotics on different types of receptors, specifically, Dopamine-2 (D2) and non-D2 receptors. However, the combination of two first-generation or two second-generation drugs is also possible. Normal precautions regarding the tolerability profile of LAI APs should be taken when choosing the drugs to be administered together.

It should be noted that, although the main guidelines suggest an integrated pharmacological and non-pharmacological treatment, previous trials with psychosocial interventions have been rarely reported in included studies. It was only in the paper by Jarosz et al. (2022) that the administration of two LAI APs was implemented after the provision of a combined treatment with individual and family therapy, psychoeducation, and rehabilitation programs [98]. In addition, Wartelsteiner et al. (2015) declared that administration of two LAI APs made it possible to participate in psychoeducational interventions [84]. Finally, it was only in the case of Mathew et al. (2018) that a previous trial with ECT was offered to patients before the prescription of two LAI APs [100]. Therefore, the results of the present review are not in line with the recommendations of the majority of guidelines, according to which psychosocial intervention should always be offered to patients with schizophrenia, especially in patients with treatment-resistant schizophrenia. In addition, assessing resistance to treatments by only considering pharmacological treatment may be a limitation of the definition of “treatment-resistance schizophrenia”. Therefore, mental health professionals should be better trained to offer non-pharmacological intervention to patients with schizophrenia, with this operating as adjunctive treatment.

It has to be noted that the patient’s willingness to take two LAI APs is rarely mentioned in the selected articles. The presence of psychotic symptoms and the emergency setting do not justify the administration of treatment without adequate consent. Adequate patient information is even more necessary when the proposed treatment is not recommended by international guidelines and involves the long-term prescription of medications. Jarosz et al. (2022) emphasize the importance of a team of doctors who can make the most appropriate decision regarding treatment monitoring the patient’s condition, while in the case proposed by Lenardon et al. (2017) an independent consultant psychiatrist authorized the treatment, given the patient’s inability to make the decision independently [96,98]. Notably, Scangos et al. (2016) repeatedly tried to obtain consent from the patient, although they did not obtain it [97]. In the majority of cases, however, a greater adherence to treatment and a greater willingness to take other oral medications after the administration of two LAI APs is reported. The achievement of a shared decision making (SDM) in clinical practice is a moral and ethical priority in the field of mental health, also considering that the available evidence reports that SMD is associated with the reduction of involuntary admission, repeated hospitalizations, total number of days of hospitalization, along with reduced overall costs of care [107,108].

In the selected articles, no new or unexpected adverse effects due to the combination of two LAIs were reported, and neither were any significant increase in side effects related to the two LAI APs that would require a treatment discontinuation reported. Therefore, according to our findings, it is possible to state that the treatment regimen with dual LAI APs is as safe as taking one long-acting drug, and does not carry an increased risk of adverse events. However, we suggest that the clinician should be prepared to counteract the onset of common side effects caused by LAI AP administration. A period of observation in a hospital setting lasting 1 to 3 h should be guaranteed, especially following the injection of olanzapine pamoate due to the risk of post-injection syndrome [109].

In the present review case reports were mainly included. Therefore, a clear picture about the efficacy and safety of two concomitant LAI APs was not possible. Further studies are needed to confirm the required efficacy and safety. In particular, these studies should include a larger sample, compare the new treatment strategies with the standard of care and should include homogeneous and standardized tools for the assessment of psychotic symptoms. Moreover, the interaction between the two administered LAI APs and between the LAI APs and the other medications prescribed should be investigated, with a focus on the efficacy of the double LAI APs prescription in the long term.

The selected studies do not fully explain the decision-making process that led to the concomitant administration of two LAI APs. Such a choice could have some advantages, such as greater patients’ compliance, greater satisfaction of the patient who does not need to take the drug daily, and greater control over the plasma concentrations of both medications. However, it has to be said some disadvantages of concomitant administration of two LAI APs also emerge, such as the discomfort deriving from two intramuscular injections with related pain at the injection site, the need to perform the injection in a hospital setting, a greater difficulty in managing non-reversible side effects and the inability to quickly change medication, whether in cases of adverse effects or loss of efficacy. Therefore, the balance between advantages and disadvantages related to the prescription of two LAI APs should be assessed in every single patient. Since all patients had previously been treated with oral and/or long-acting antipsychotics, it can be suggested, on the basis of our data, that the administration of two LAI APs should be reserved for patients who have not been shown to previously benefit from adequate antipsychotic treatment. In particular, the results of the present study highlight that LAI APs should be reserved to patients who have not been shown to benefit from previous adequate antipsychotic treatment, and had a poor compliance with treatments, especially when other psychosocial and pharmacological strategies failed.

### Limitations

The present review has some limitations which must be acknowledged. First, the search strategy has been limited to schizophrenia and related disorders; in fact, no comparison has been made between the different disorders of the psychotic spectrum nor other disorders treated—even off-label—with LAI APs, such as bipolar disorder and personality disorders—neither has been taken into account [110,111,112]. Another limitation is due to the extreme heterogeneity of the samples and of the assessment tools used for measuring treatment efficacy. This could be due the complexity of the psychotic symptoms, including different symptomatologic dimensions such as positive, negative, affective, aggressive, and cognitive symptoms. Another reason for this heterogeneity could be the absence of randomized controlled clinical trials on the topic. Therefore, it is not possible to draw certain conclusions. Third, in the literature there are mainly case reports and case series describing the use of double LAI Aps, which exposes a selection bias, since the authors may have a tendency to only report extreme cases. Moreover, we included in this review the study by Kenar et al. (2023), which included 12 patients not diagnosed with schizophrenia or other psychotic disorders (out of 83 recruited patients) [95]. Unfortunately, it was not possible to extrapolate data on patients with psychosis. However, it was not appropriate to exclude this study from the present review, given its large sample size and observational design. Lastly, this revision does not take into account the costs related to the use of two LAI APs in the same patient; long-acting APs tend to be more expensive than oral formulations, and the costs of the medical personnel dedicated to the administration and patients’ observation should be added.

## 5. Conclusions

The findings of the present systematic review clearly highlight the complexity of the management of schizophrenia and related disorders, while identifying the concomitant administration of two antipsychotics in long-acting injectable formulation as a promising, effective, and relatively safe therapeutic strategy that combines antipsychotic efficacy with a good tolerability profile, especially in TRS and those cases where oral medications are refused or not adequately taken.

Reducing the number and duration of hospitalizations in psychiatric settings, increasing recovery, and improving patients’ quality of life could offset the high costs of LAI therapy. However, despite its wide use in clinical practice, it is not possible to draw definitive conclusions on the “best practice” for the use of dual LAI Aps, due to the lack of solid evidence. Therefore, further studies are needed to evaluate the risk/benefit ratio of this therapeutic regimen, which feature a larger and more homogeneous sample and standardized assessment tools, as this will help to update national and international guidelines by including the possibility of administering two concomitant LAI APs in patients with schizophrenia and related disorders.

## Figures and Tables

**Figure 1 brainsci-14-00433-f001:**
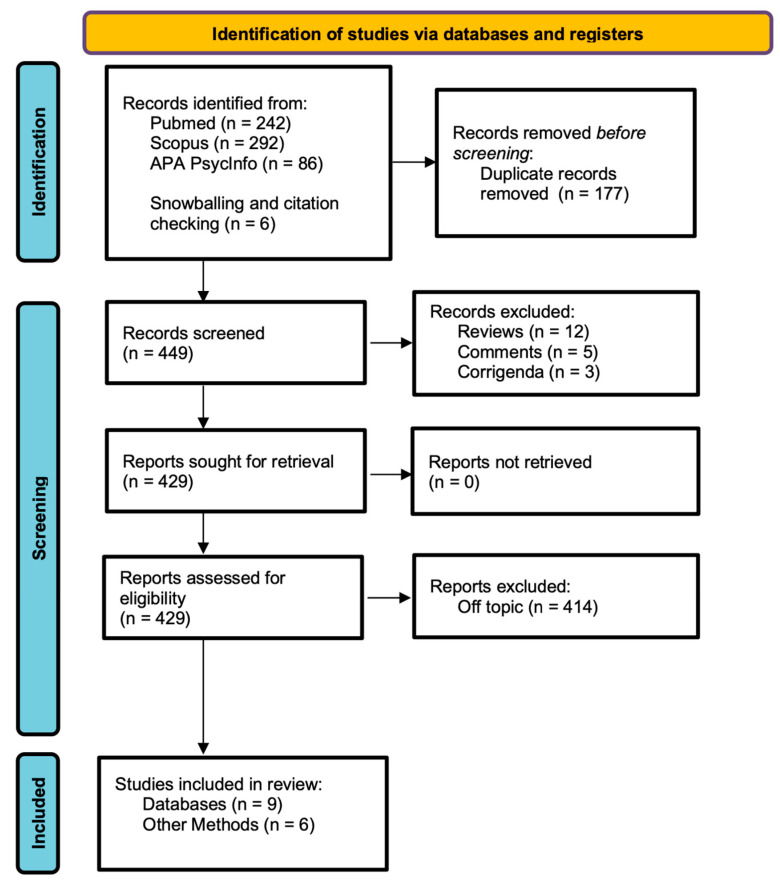
Flowchart of the selection process, according to PRISMA guidelines.

**Table 1 brainsci-14-00433-t001:** Risk of bias assessment in non-randomized studies of intervention (NRSI).

First Author (Year of Publication)	Type of Study	Pre-Intervention Domains		At-Intervention Domain	Post-Intervention Domains				Overall Risk of Bias
		Confounding Bias	Selection Bias	Information Bias	Confounding Bias	Selection Bias	Information Bias	Reporting Bias	
Calvin N et al. (2023) [94]	Observational retrospective study	Moderate	Low	Low	Moderate	Moderate	Moderate	High	High
Kenar ANI et al. (2023) [95]	Observational retrospective study	High	High	Low	Moderate	Moderate	High	High	High

**Table 2 brainsci-14-00433-t002:** Summary of selected clinical studies regarding the concomitant administration of two LAI APs.

First Author (Year of Publication) Type of Study, Country	Sample’s Characteristics	Previous APs Trials	Dual LAI APs Schedule(s)	Outcome Assessment	Results
Calvin N et al. (2023) [94] Observational retrospective study, France	N = 13 (10M; 3F); mean age 24.5 ± 23.9 9 SCZ; 4 SCA	Mean of 5.5 ± 2.3 different APs Three cases received CLO, suspended due to non-adherence, sedation, weight gain	A total number of six different combinations has been used: 3 × dZUC + dHAL; 3 × dZUC + dARI; 3 × dZUC + dPAL; 2 × dARI + dHAL; 1 × dARI + dPAL; 1 × dPIP + dFLP	Efficacy measures: hospitalization (number and length) Tolerability measures: BMI, agranulocytosis, lipid profile, sugar levels	Significant differences between 12 months pre-dual LAI APs and 12-month post-dual LAI APs were found. The number of hospitalizations decreased by 50% (2.6 ± 1.3 vs. 1.3 ± 1.3) and the length of hospitalizations decreased by 33% (142 d ± 79 vs. 95 d ± 94).No significant differences observed in tolerance outcomes.
Kenar ANI et al. (2023) [95] Observational retrospective study, Turkey	N= 83; mean age 34.7 ± 9.5 37 SCZ; 34 SCA; 12 Other psychiatric disorder	96.4% received oral medications before dual LAI APs	A total number of 11 different combinations has been used: 26 × dPAL + dARI; 20 × dPAL + dFLU; 15 × dPAL + dZUC; 5 × dPAL + dHAL; 5 × dFLU + dHAL; 3 × dFLU + dRIS; 3 × dZUC + dFLU; 2 × dPAL/3 m + dARI; 2 × dZUC + dHAL; 1 × dRIS + dHAL; 1 × dZUC + dARI	Hospitalization	The average number of hospitalizations was significantly lower in the post-dual LAI APs period (5.95 vs. 0.99, *p* < 0.001). The number of patients did not require hospitalization increased post-dual LAI APs (9 vs. 39, *p* < 0.001). No significant adverse effects related to the use of dual long-acting injectable drugs were observed.

AP: Antipsychotic; dARI: Aripiprazole depot; BMI: Body Mass Index; CLO: Clozapine; dFLU: Flupentixol depot; dFLP: Fluphenazine depot; dHAL: Haloperidol depot; LAI: Long-acting injectable; m: month(s); dOLA: Olanzapine depot; dPAL: Paliperidone depot; dPIP: Pipotiazine depot; dRIS: Risperidone depot; SCA: Schizoaffective disorder; SCZ: Schizophrenia; dZUC: Zuclopenthixol depot.

**Table 3 brainsci-14-00433-t003:** Summary of selected studies reporting case reports or case series regarding the concomitant administration of two LAI APs.

First Author (Year of Publication), Country Report’s Characteristics (Sex, Age, Main Diagnosis, Duration of the Illness, Reported Comorbidities)	Previous APs Trials/Previous Trial of Clozapine (Reason for Discontinuation)	Pre-LAI APs Symptoms/Psychiatric Assessment	Dual LAI APs Schedule	AEs Emerged after Double LAI AP	Results
Alba P (2017) [87], Spain ^‡^					
Male, 32; SCZ (11 y) N/A	CLO, OLA, RIS, dRIS, HAL, dRIS + HAL, ARI	N/A	dRIS + dHAL	N/A	Symptoms stabilization was achieved with two LAI APs.
	YES (leukocytopenia)	Clinical evaluation			
Jarosz M et al. (2022) [98], Poland					
Male, 34; SCZ (12 y) Ob; HLD; acne inversa;	dPER; RIS + AMI; QUE; dOLA; dOLA 300 mg/2 w + HAL 15 mg/d; ZUC 75 mg/d	Delusions; disorganized speech; social withdrawal; blunted affect; avolition; aggressive behavior	dOLA 300 mg/2 w + dZUC 200 mg/2 w (alternately on a weekly schedule)	No	After dual LAI APs administration, positive and negative symptoms improved and the total PANSS score reduced from 90 to 57.
	NO (non-adherence)	PANSS			
Ladds B et al. (2009) [89], USA					
Woman, 49; SCZ N/A	RIS; RIS + FLP	Delusion; aggressive and bizarre behavior; disorganized speech; poor compliance; low insight	dFLP + dRIS	No	Clinical response to two oral APs was confirmed after transition to depot formulation of both medications.
	N/A	Clinical evaluation			
Legrand G et al. (2014) [88], France					
Male, 51; SCA (32 y) N/A	HAL 30 mg/d; RIS 4 mg/d; LEV 120 mg/d; QUE 800 mg/d; CLO	Delusion; hallucinations; agitation; avolition; expansive mood; disorganized speech	dPAL 100 mg/4 w + dOLA 300 mg/4 w (alternately on a biweekly schedule)	No	After three months, all the scores of CRDPSS items decreased; two more months after, the patient’s psychiatric symptoms were stable.
	YES (N/A)	CRDPSS			
Lenardon A et al. (2017) [96], UK					
Male, 52; SCZ (28 y) N/A	CLO, ECT, dOLA + ARI	Disorganized/bizarre behavior; violence; hostility; poor grooming; poor compliance	dARI 400 mg/m + dOLA 405 mg/10 d	No	At the 18-month follow-up: PANSS-EC reduced from 102 to 73; BPRS score reduced from 81 to 47; CGI Illness Severity Score reduced from seven to three; CGI Global Improvement reduced from seven to two.
	YES (non-adherence)	BPRS; PANSS-EC; CGI			
Li FL et al. (2018) [99], Taiwan					
Woman, 35; SCZ (10 y) N/A	AMI 800 mg/d; RIS 6 mg/d; OLA 20 mg/d; dFLU 40 mg/2 w	Delusions; hostility; lability	dARI 400 mg/4 w + dFLU 20 mg/4 w (alternately on a biweekly schedule)	N/A	Symptoms stabilization was achieved.
	N/A	Clinical evaluation			
Male, 46; SCZ (20 y) N/A	QUE 800 mg/d; OLA 20 mg/d; HAL 20 mg/d	N/A	dFLU 40 mg/4 w + dPAL 150 mg/4 w (alternately on a biweekly schedule)	No	Psychotic symptoms improved significantly.
	N/A	Clinical evaluation			
Mathew C et al. (2018) [100], Australia					
4 males, 1 female; age range 36–64; SCZ (3–17 y) N/A	CLO; combinations of oral and LAI APs; ECT	N/A	2 × dOLA + dARI; 3 × dZUC + dOLA/dRIS (alternately on a weekly/biweekly schedule)	N/A	The total number of hospitalizations experienced by patients reduced from 40 to only one hospitalization lasting four weeks in the post-LAI APs phase.
	YES (non-adherence; no response; AEs)	Hospitalization			
McInnis P et al. (2019) [101], Australia					
Male, 17; SCZ (first episode) SUD	dOLA 405 mg/2 w	Delusions (persecutory, Capgras); hallucinations; violence	dOLA 405 mg/2 w + dZUC 600 mg/2 w	Mild bradykinesia	Positive symptoms resolved, but blunt affect and avolition appeared; BPRS scores improvement observed in the next five-month follow-up.
	NO (N/A)	BPRS			
Male, 17; SCZ (3 y) SUD; Conduct disorder	dPAL 100 mg/m; dPAL 150 mg/m + OLA 40 mg/d; dZUC 400 mg/2 w	Delusions (persecutory); hallucinations; violence	dPAL 150 mg/m + dZUC 400 mg/2 w	Polydipsia; polyuria	Symptoms, behavior and BPRS scores improved in the next two-month follow-up. Occasional responding to funny voices.
	NO (refusal of oral medication)	BPRS			
Male, 16; SCZ (first episode) SUD	dOLA; dPAL 150 mg/m	Disorganized/ bizarre behavior; avolition; social withdrawal; violence	dPAL 150 mg/m + ZUC 600 mg/15 d	No	Persecutory delusions and negative symptoms gradually lessened over the four-month follow-up time, as well as BPRS score. All patients showed a significant clinical improvement after dual LAI APs, with a mean BPRS reduction from 89 to 44 (*p*= 0.005).
	NO (refusal of oral medication)	BPRS			
Ross C et al. (2013) [86], USA					
Male, 44; SCZ (10 y) Ob; AH; HLD	dHAL 400 mg/2 w + HAL 20 mg/d	Delusion (paranoid); poor compliance	dHAL 400 mg/2 w + dPAL 156 mg/4 w	No	Two LAI APs regimen favored the patient’s discharge after four months of acute hospitalization, with only minor residual symptom.
	N/A	Clinical evaluation			
Scangos KW et al. (2016) [97], USA					
Male, 26; SCZ N/A	dPAL 156 mg/m; PAL 156 mg/m + OLA 30 mg/d; CLO 350 mg/d; CLO 600 mg/d + HAL; HAL 10 mg/d + OLA 20 mg/d; dHAL 50 mg/m	Delusions (paranoid); hallucinations; disorganized speech and behavior; agitation; impulsiveness; violence	dOLA 405 mg/m + dHAL 50 mg/m	N/A	Patient’s compliance to oral medications and psychotic symptoms improved, with only sporadic delusion.
	YES (refusal of oral medication)	Clinical evaluation			
Wartelsteiner F et al. (2015) [84], Austria					
Male, 30; SCZ N/A	OLA 30 mg/d; RIS 8 mg/d; CLO 250 mg/d; dFLP 25 mg; dFLU 20 mg/2 w; ZIP 80 mg/d; QUE 1200 mg/d; SER 16 mg/d; ARI 30 mg/d; HAL 6 mg/d; dRIS 50 mg/2 w + RIS 8 mg/d; dHAL 150 mg; dRIS 50 mg/2 w + OLA 30 mg/d; dOLA 300 mg/2 w + RIS 3 mg/d	Delusions (persecutory); hallucinations; disorganized speech; distortion in language; self-injurious behaviors; poor compliance	dOLA 300 mg/2 w + dRIS 100 mg/2 w	No	Dual LAI APs regimen determined symptoms remission, social reintegration, and higher quality of life.
	YES (sedation, orthostatic hypotension, leukocytopenia)	Clinical evaluation			
Yazdi K et al. (2014) [90], Austria ^†^					
Male, 44; SCA (21 y) N/A	FLP; dFLP; HAL; dHAL; RIS; RISd; ZUC, dZUC; CLO; OLA	Delusion; aggressive behavior	dRIS 50 mg/2 w + dZUC 500 mg/2 w (simultaneously)	No	Although complete remission did not occur, sufficient stabilization was achieved.
	YES (leukocytopenia)	Clinical evaluation			
Youykheang M et al. (2023) [85], Canada					
Male, 51; SCZ (5 y) SUD	dPAL 150 mg + LUR 100 mg/d; dZUC 200 mg	N/A	dZUC 150 mg/2 w + dARI 400 mg/4 w	EPS	Hospitalizations: 2 in 5 y –> 0 in 28 m EDV: 0 in 5 y –> 3 in 28 m
	NO (refusal)	Hospitalization; EDV			
Male, 49; SCA (21 y) SUD; PD	dFLU 120 mg; OLA 35 mg/d; LOX 300 mg/d	N/A	dPAL 150 mg/3 w + dFLU 150 mg/3 w	N/A	Hospitalizations: 10 in 10 y –> 0 in 30 m EDV: 9 in 10 y –> 3 in 30 m
	NO (N/A)	Hospitalization; EDV			
Woman, 32; SCZ (12 y) PD; eating disorder unspecified	OLA 20 mg; dARI 400 mg; OLA 10 mg/d + dPAL 150 mg; dZUC 75 mg/2 w; CLO 225 mg/d	N/A	dPAL 150 mg/4 w + dARI 400 mg/4 w (simultaneously)	N/A	Hospitalizations: 12 in 3 y –> 2 in 21 m EDV: 21 in 3 y –> 3 in 21 m
	YES (refusal)	Hospitalization; EDV			
Male, 48; SCA (31 y) SUD; WPW	RIS 2 mg/d; CLO 300 mg/d; dARI 400 mg; dPAL 150 mg; OLA 15 mg/d; dFLU 125 mg; LOX 25 mg/d; QUE 700 mg/d; HAL 10 mg/d; TRI 50 mg/d	N/A	dARI 150 mg/4 w + dPAL 150 mg/4 w (simultaneously)	N/A	Hospitalizations: 11 in 31 y –> 0 in 45 m EDV: 9 in 31 y –> 0 in 45 m
	YES (sedation)	Hospitalization; EDV			
Woman, 67; SCZ (>20 y) HA	OLA 10 mg/d; dZUC 200 mg; dPAL 100 mg	N/A	dARI 400 mg/3 w + dZUC 200 mg/3 w (simultaneously)	EPS	Hospitalizations: 3 in 2 y –> 2 in 33 m EDV: 0 in 2 y –> 8 in 33 m
	NO (refusal of oral medication)	Hospitalization; EDV			
Woman, 37; SCA (10 y) PD	QUE 500 mg/d; LOX 25 mg/d; dZUC 150 mg/2 w; dPAL 150 mg/3 w; CLO	N/A	dARI 400 mg/3 w + dPAL 150 mg/3 w (simultaneously)	N/A	Hospitalizations: 33 in 10 y –> 2 in 43 m EDV: 0 in 10 y –> 54 in 43 m
	YES (refusal)	Hospitalization; EDV			
Woman, 37; SCZ (2 y) Borderline IQ	RIS 3 mg/d; ARI 25 mg/d; OLA 2,5 mg/d	N/A	dARI 400 mg/4 w + dPAL 150 mg/4 w (simultaneously)	N/A	Hospitalizations: 2 in 3 y –> 0 in 16 m EDV: 1 in 3 y –> 0 in 16 m
	NO (N/A)	Hospitalization; EDV			
Woman, 55; SCZ (>20 y) None	ARI 20 mg/d; RIS 2 mg/d	N/A	dARI 400 mg/4 w + dPAL 100 mg/4 w (simultaneously)	N/A	Hospitalizations: 1 in 4 m –> 0 in 12 m EDV: 0 in 4 m –> 0 in 12 m
	NO (N/A)	Hospitalization; EDV			

^†^ Article in German; ^‡^ Article in Spanish. AEs: Adverse events; AH: Arterial hypertension; AMI: Amisulpiride; AP: Antipsychotic; ARI: Aripiprazole; BPRS: Brief Psychiatric Rating Scale; CGI: Clinical Global Impression; CLO: Clozapine; CRDPSS: Clinician-Rated Dimensions of Psychosis Symptom Severity; d: day; dARI: Aripiprazole depot; dFLP: Fluphenazine depot; dFLU: Flupentixol depot; dHAL: Haloperidol depot; dRIS: Risperidone depot; dZUC: Zuclopenthixol depot; ECT: Electroconvulsive Therapy; EDV: Emergency Department Visit; EPS: Extrapyramidal Symptoms; FLP: Fluphenazine; HAL: Haloperidol; HLD: Hyperlipidemia; LAI: Long-Acting Injectable; LEV: Levomepromazine; LOX: Loxapine; LUR: Lurasidone; m: month(s); N/A: Not Available; Ob: Obesity; OLA/dOLA: Olanzapine/Olanzapine depot; PAL/dPAL: Paliperidone/Paliperidone depot; PANSS: Positive and Negative Symptoms Scale; PANSS-EC: Positive and Negative Symptoms Scale-Excited Component; dPER: Perphenazine depot; PD: Personality Disorder; QUE: Quetiapine; RIS: Risperidone; SCA: Schizoaffective Disorder; SCZ: Schizophrenia; SER: Sertindole; SUD: Substance Use Disorder; TRI: Trifluoperazine; TRS: Treatment Resistant Schizophrenia; w: week; WPW: Wolf–Parkinson–White syndrome; y: year; ZIP: Ziprasidone; ZUC: Zuclopenthixol.

**Table 4 brainsci-14-00433-t004:** List of symptoms reported before the introduction of dual LAI APs. The number of times the symptom is cited in the selected articles is indicated in brackets.

Positive Symptoms	Negative Symptoms	Mood Symptoms	Aggressive Behavior
Delusions (10) Hallucinations (5) Disorganized speech (4) Disorganized/bizarre behavior (3) Agitation (2) Distortion in language (2)	Avolition (3) Social withdrawal (2) Blunted affect (1) Poor grooming (1)	Expansive mood (1) Lability (1)	Violence (5) Hostility (2) Aggressive behavior (3) Impulsiveness (1) Self-injurious behaviors (1)

**Table 5 brainsci-14-00433-t005:** Different combination of long-acting antipsychotics used in the selected studies.

		First Generation Antipsychotics					Second Generation Antipsychotics				
		Haloperidol Decanoate	Zuclopenthixol Decanoate	Pipotiazine Decanoate	Fluphenazine Enanthate	Flupentixol Decanoate	Aripiprazole Monohydrate	Olanzapine Pamoate Monohydrate	Risperidone ISM	Paliperidone Palmitate	Paliperidone Palmitate 3M
First generation antipsychotics	Haloperidol decanoate										
	Zuclopenthixol decanoate	**5** 3 × Calvin 2 × Kenar	**FGA + FGA**								
	Pipotiazine decanoate										
	Fluphenazine enanthate			**1** 1 × Calvin							
	Flupentixol decanoate	**5** 5 × Kenar	**3** 3 × Kenar								
						**FGA + SGA**					
Second generation antipsychotics	Aripiprazole monohydrate	**2** 2 × Calvin	**6** 3 × Calvin 1 × Kenar 2 × Youykheang			**1** 1 × Li					
	Olanzapine pamoate monohydrate	**1** 1 × Scangos	**3** 1 × Mathew * 2 × McInnis				**3** 1 × Lenardon 2 × Mathew	**SGA + SGA**			
	Risperidone ISM	**2** 1 × Kenar 1 × Alba	**2** 1 × Yazdi 1 × Mathew *		**1** 1 × Ladds	**3** 3 × Kenar		**1** 1 × Wartelsteiner			
	Paliperidone palmitate 1M	**6** 5 × Kenar 1 × Ross	**20** 3 × Calvin 15 × Kenar 2 × McInnis			**22** 20 × Kenar 1 × Li 1 × Youykheang	**32** 1 × Calvin 26 × Kenar 5 × Youykheang	**1** 1 × Legrand			
	Paliperidone palmitate 3M						**2** 2 × Kenar				

* Only two of the three patients reported in Matew (2018) [100] are listed in the table, since it is not possible to determine if the third patient received zuclopenthixol decanoate + risperidone ISM or zuclopenthixol decanoate + olanzapine pamoate.

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
