# Peer review of "Combination of Two Long-Acting Antipsychotics in Schizophrenia Spectrum Disorders: A Systematic Review"

_brainsci, 2024, doi:10.3390/brainsci14050433_

Round 1

Reviewer 1 Report

Comments and Suggestions for Authors

Thank you for giving me the opportunity to review this manuscript.

1) Please attach the PRISMA checklist and fill in the page numbers.

2) I think this manuscript was so interesting, but it is difficult to explain whether dual long-acting antipsychotics were safe and effective in treatment-resistant Schizophrenia. Please expain what is the disadvantage of assessing safety and efficacy of dual long-acting antipsychotics in previous studies and what kind of feasible studies are warranted. 

3) It is better to explain what is the advantage and disadvantage of dual oral antipsychotics from previous studies. Moreover, it is bettter to show what is the differece between dual oral antipsychotics and dual long-acting antipsychotics, and to discuss what kind of population is best to use dual long-acting antipsychotics.

I think it is necessary to revise the manuscript.

2) 

Reviewer 2 Report

Comments and Suggestions for Authors

Author has done a good attempt however they are too much biologically oriented 

Intro: 

Schizophrenia Spectrum > seems to be influenced too much by recent DSM but it does not conform to real Krapelinian Schizophrenia 

Modern world Schizophrenia is diluted , and  mixed with drug, personality etc. 

7 percent mortality > is it same after recent medicined discovery , if not then is there a point of this new drugs 

Antipsychotic mainstay of treatment as mentioned in paper > is it really true , can author look at NICE guidelines , please also include psychosocial treatment written by Joanna Moncriff 

Might not be main , what about TMS, Psychedelics, ECT > need to provide it too 

Table - needs sample size, need for double depot

Discussion 

Seems fine 

However needs to explain all the neurobiology in all the combination 

Whether its a random choice or biologically informed one 

No where its mentioned pn psychosocial , if tried before choosing two Depot 

if CBT or dynamic therapy tried 

Schema therapy tried or not 

Conclusion: Recovery is possible without remission 

Author is wrong , rather not updated with recent development 

Please read and incorporate the CHIME model of recovery 

The paper needs to incorporate the social and psychological views too 

what about the side effects of two depot 

Do they agree or its forced on them in the studies 

Human rights violation the the studies chosen 

Comments on the Quality of English Language

as before 

needs major review 

Reviewer 3 Report

Comments and Suggestions for Authors

The topic is of interest and practical. The Search Strategy and analyzed articles are acceptable. Although the manuscript needs revisions before next round of review:

1. 'of the literature' is not helpful in the title.

2. Do you mean resistant to monotherapy pharmacological treatment of first sentence of Abstract and in Introduction?

3. Search Strategy should be described more in Abstract.

4. Discussion should be replaced with Conclusions in Abstract.

5. 'Although schizophrenia is no longer considered a deteriorating progressive disease ' should be softened.

6. LAIAPs and their variety and availability should be discussed briefly.

7. Key-words could be transferred to a tale as a supplementary file.

8. Figures are not clear enough.

9. Tables should be ordered based on first author`s family name, alphabetically, or year of study.

10. I believe that patients without Schizophrenia/ schizoaffective diagnosis should be removed from the analyses and report.

11. 'Mood symptoms' is duplicated item in Table-4. In addition, 'Cognitive
symptoms' should be removed from this table.

12. Legend of Table-5 should be more clearly informative.

13. It seems that 'mid ' should be replaced with 'mild' (line 289).

14. Conclusions should be extremely summarized and emphasized on the findings of the current study with scare background and without citation.

Comments on the Quality of English Language

Minor editing of English language required

Round 2

Reviewer 2 Report

Comments and Suggestions for Authors

Accept 

Author Response

R #2: Accept

Thank you again for reviewing our manuscript.

Reviewer 3 Report

Comments and Suggestions for Authors

Thank you for revisions. It has been improved substantially. Although, a few further revisions are needed still:

1. I can`t see 'Conclusion' sub-section in Abstract of revised manuscript.

2. 'and Transcranial Magnetic Stimulation (TMS)' should be removed from lines 77-8.

3. Limited availability of LAAPs, particularly SGs, in LMICs should be discussed briefly.

4. In my opinion, Conclusion should be summarized again and, particularly, first paragraph could be even removed. Any citation in Conclusion should be avoided. in addition, main findings of the current study should be emphasized.
